# The Antiangiogenic Effect of VEGF-A siRNA-FAM-Loaded Exosomes

**DOI:** 10.3390/bioengineering12090919

**Published:** 2025-08-26

**Authors:** Woojune Hur, Basanta Bhujel, Seheon Oh, Seorin Lee, Ho Seok Chung, Jin Hyoung Park, Jae Yong Kim

**Affiliations:** 1Department of Ophthalmology, Asan Medical Center, University of Ulsan College of Medicine, Seoul 05505, Republic of Korea; dnwnsgj@naver.com (W.H.); basantabhujel86@gmail.com (B.B.); sh.oh@amc.seoul.kr (S.O.); ra02582@amc.seoul.kr (S.L.); chunghs@amc.seoul.kr (H.S.C.); 2Department of Medical Science, University of Ulsan Graduate School, Seoul 05505, Republic of Korea; 3MS Eye Clinic, Seoul 06512, Republic of Korea; drpark99@naver.com

**Keywords:** VEGFA siRNA-loaded exosomes, neovascularization, exosomes, apoptosis, tube formation

## Abstract

Neovascular ocular diseases are caused by vascular endothelial growth factor A (VEGFA) overexpression. Thus, VEGFA inhibition is considered the main strategy for treating ocular neovascularization. However, existing anti-VEGF therapies have several limitations in stability and delivery efficiency. To overcome the limitations, exosome-based VEGF siRNA delivery technology has attracted attention since exosomes have the advantages of high in vivo stability and excellent intracellular delivery efficiency. Additionally, loading VEGFA siRNA into exosomes not only allows for targeting specific cells or tissues but can also improve therapeutic efficacy. Our research team purified and concentrated exosomes using chromatography techniques, added fluorescein amidite (FAM)-labeled VEGFA siRNA into exosomes, and observed the novel effect of drug delivery in vitro. This study successfully introduced hVEGFA siRNA-FAM into target cells, with high efficacy particularly at 48 h after treatment. Furthermore, the enhanced inhibition of VEGFA expression at 48 h post-treatment was confirmed. FACS analysis was performed using the apoptosis markers Annexin V-FITC (green) and PI-PE (red) to confirm the presence or absence of apoptosis. Both groups treated with hVEGFA siRNA-FAM-EXO (1) and hVEGFA siRNA-FAM-EXO (2) showed increased apoptosis as the exposure time passed compared to the untreated group (0 h). hVEGFA siRNA-FAM-EXO treatment effectively induced apoptosis. After 24 h, early apoptosis was 12.9% and 13.9% and late apoptosis was 1.5% and 3.7% in hVEGFA siRNA-FAM-EXO groups (1) and (2), respectively. After 48 h, early apoptosis was 23.9% and late apoptosis was 39.4% and 17.8% in hVEGFA siRNA-FAM-EXO groups (1) and (2), respectively, indicating a time-dependent pattern of apoptosis progression. Additionally, tube formation of human vascular endothelial cells (HUVECs) was induced to confirm the effect of VEGFA siRNA-loaded exosomes on the angiogenesis assay in vitro. Compared with controls, angiogenesis became significantly weakened in hVEGFA siRNA-FAM-EXO (1)- and hVEGFA siRNA-FAM-EXO (2)-treated groups at 48 h post-treatment and completely disappeared at 72 h, probably occurring due to decreased VEGFA, PIGF, and VEGFC in the intracellular cytosol and conditioned media secreted by VEGFA siRNA-FAM in HUVECs. In conclusions, FAM-tagged VEGFA siRNA was packed into exosomes and degraded over time after tube formation, leading to cell death due to a decrease in VEGFA, PIGF, and VEGFC levels. This study is expected to support the development of in vivo neovascularization models (keratitis, conjunctivitis, or diabetic retinopathy models) in the future.

## 1. Introduction

In the eye, neovascularization causes central vision loss in patients with pathological myopia, representing the formation of new vessels from existing blood vessels [1]. Neovascularization is mainly observed in myopic choroiditis, diabetic retinopathy, macular degeneration, retinal vein occlusion, and corneal neovascularization [2,3,4,5]. These vision-impairing diseases are mainly treated with anti-vascular endothelial growth factor (VEGF) medications, which inhibit neovascularization and are administered directly into the vitreous through an injection. VEGFA is involved in angiogenesis, lymphangiogenesis, and endothelial cell growth, inducing endothelial cell proliferation, promoting cell migration, inhibiting apoptosis, and increasing vascular permeability [6]. Furthermore, VEGFA has been associated with abnormal angiogenesis in various diseases, including cancer, diabetic retinopathy, and macular degeneration [7]. Anti-VEGF treatment is effective in patients with diabetic retinopathy and macular degeneration [8].

Anti-VEGF therapy prevents or restores vision loss by inhibiting the action of VEGF proteins that are abnormally produced in the eye due to retinal diseases, such as macular degeneration or diabetic macular edema, thereby preventing new blood vessel formation and blood leakage [9]. This approach has become a standard treatment for various retinal diseases. The main anti-VEGF drugs include ranibizumab (Lucentis), a recombinant VEGF antibody fragment, and bevacizumab (Avastin), which was originally used to treat cancer but repurposed for ocular use. Both drugs have shown similar efficacy and safety in clinical practice [10]. Anti-VEGF therapy is mainly administered through intravitreal injection, which is one of the most widely performed procedures in ophthalmology. Although intraocular injections are highly safe with a low risk of complications when performed by experienced medical staff, patients feel a psychological burden about receiving injections directly into the eye, and there are also concerns about vision-threatening complications that may occur after the injection [11]. In addition, standard treatment for the disease is anti-VEGF antibody injection therapy. This treatment suppresses neovascularization by neutralizing the already produced VEGF protein; however, it has limitations such as a short half-life and the need for repeated injections [8]. Furthermore, not all patients respond satisfactorily to anti-VEGF therapy. Some patients show limitations in their ability to respond to initial treatment (intrinsic resistance) or in their ability to respond over time (acquired resistance). These findings highlight the need for improvements in existing anti-VEGF therapies and new treatment strategies [9,12]. In some cases, steroid injections are used concurrently to reduce inflammation and swelling; however, steroid use can cause various ophthalmic side effects, including increased ocular pressure, glaucoma, cataract formation, delayed wound healing, and increased risk of infectious diseases [13,14]. In contrast, the siRNA-based strategy investigated in this study fundamentally suppresses the production of the VEGF protein itself, thereby enabling more sustained and efficient neovascularization suppression than existing antibody treatments [15]. This can contribute to long-term vision improvement and reduce the need for frequent injections. In addition, exosomes stably protect nucleic acid therapeutics such as siRNA and enable stable circulation in vivo, maximizing drug delivery efficiency and therapeutic efficacy [16,17].

Exosomes are very small (30–150 nm) extracellularly secreted vesicles derived from cells, originating from the cell membrane and being released outside the cell [18,19]. They contain various biologically active substances, including proteins, lipids, and RNA (microRNA, mRNA, tRNA, etc.) [20], and function as key mediators of intercellular signaling and material transport [21,22]. By delivering these substances to other cells, exosomes regulate various physiological functions such as immune responses, cell growth, and differentiation [23]. Recent studies highlight several important advantages of exosomes as a next-generation drug delivery system. First, high drug delivery efficiency: Exosomes have a lipid bilayer structure similar to biological membranes, so they can be naturally absorbed or fused with target cells, which greatly improves the efficiency of intracellular introduction of various therapeutic substances [24]. Second, highly safe: Since exosomes are derived from autologous cells, they induce fewer immune responses compared to foreign substances, which reduces the risk of immunogenicity and toxicity when delivering therapeutic substances or genes, thereby increasing safety [25]. In addition, they tend to be relatively stable in the body and are resistant to enzymatic degradation, allowing for prolonged therapeutic effects [26]. Third, the possibility of transporting various types of contents: Exosomes can effectively encapsulate and transport a wide range of bioactive substances, ranging from small molecular compounds to proteins and nucleic acids (siRNA, miRNA, etc.), making them suitable for treating a wide range of diseases [27]. Fourth, high targeting ability: Specific proteins or lipids present on the surface of exosomes have a natural affinity for specific cells or tissues, facilitating precise drug delivery to desired target sites, reducing non-target effects, and maximizing therapeutic efficacy [21]. These unique properties position exosomes as a promising platform for clinical drug delivery.

Our hypothesis is that exosome-loaded VEGFA siRNA inhibits angiogenesis by reducing the expression of VEGFA mRNA and several other proteins, including VEGFA. Furthermore, if exosomes carry VEGFA siRNA, they may provide higher safety and efficacy than existing treatments.

## 2. Materials and Methods

### 2.1. Cell Culture

Human umbilical cord-derived mesenchymal stem cells (MSC; PCS-500-010, ATCC, Manassas, VA, USA) and human umbilical vein-derived endothelial cells (HUVECs; PCS-100-010, ATCC) were used in the experiment. MSCs were mixed with Minimum Essential Medium-Alpha (MEM Alpha) (12571-063, Gibco, Inc., Waltham, MA, USA) and 10% fetal bovine serum (FBS, 10099-141, Gibco, Inc.) and cultured in an incubator supplied with 5% CO_2_, 95% air, and 37 °C with 1% antibiotic–antimycotic (15240-062, Gibco, Inc.). For HUVECs, human large-vessel endothelial cell basal medium (Medium 200; M200500, Gibco, Inc.), low-serum-growth supplement (LSGS; S00310, Gibco, Inc.), and 10% FBS (10099-141, Gibco, Inc.) were mixed and cultured under the same conditions as MSCs using 1% antibiotic–antimycotic (15240-062, Gibco, Inc.).

### 2.2. Exosome Purification

Exosomes were purified using multicolumn chromatography. The method was described previously [28]. The simple method of pure separation is as follows. First, MSCs were cultured in MEM-Alpha medium (150 cm^2^ cell culture dish) without FBS for 5 days to prepare conditioned medium. Conditioned medium without cells was collected on the day of collection.

As a sample, the collected cell medium was centrifuged at 10,000× *g* for 30 min at 4 °C to remove the pellet containing dead cells and debris. After that, only the supernatant was carefully separated and prepared as the final sample. A total volume of 0.5 mL of the prepared sample was loaded onto the chromatography column. Once the entire sample had completely permeated the first disk of the column, phosphate-buffered saline (PBS) was added to the top of the column to initiate elution. At this time, the flowing eluate was collected in 0.5 mL fractions. Starting with the first, a total of 13 fractions were collected by sequentially replacing the microtube. Among them, fractions 11 and 12, which were rich in exosomes, were collected as the final exosome suspension. The collected exosome suspension was concentrated using a centrifugal filter (Amicon Ultra-0.5 mL; Millipore, Inc., Burlington, MA, USA), yielding a final volume of 100 μL of highly concentrated exosomes.

### 2.3. Transmission Electron Microscopy

A transmission electron microscope (TEM; HT7800, Hitachi, Ltd., Tokyo, Japan) was used to obtain exosome electron microscopy images. The TEM used a tungsten hairpin filament, a hot electron gun with DC heating, an acceleration voltage of 100 kV (up to 120 kV), an imaging camera of AMT nanosprint, and a resolution of 0.20 nm (lattice imaging at 100 kV). The magnification, utilizing an optional LN2 trap (cold finger), was set at ×200–×200,000 in a high contrast mode and ×4000–×600,000 in a high-resolution mode.

A negative staining protocol was used to obtain exosome TEM images. Specifically, a glow discharge coater (SMC12R-Plus; Semian, Inc., Daejeon, Republic of Korea) was used to make the grid hydrophilic. Second, the grid surface was sprayed with glow discharge for 10–15 s, followed by dropping a drop of the sample. Afterward, the remaining solution was removed with filter paper. Then, a drop of 1% uranyl acetate solution was sprayed, and the remaining solution was removed with filter paper. Finally, drying was performed.

### 2.4. Nanoparticle Tracking Analysis

Nanoparticle tracking analysis (NTA; Malvern NanoSight, Inc., LM10, Malvern, UK) was performed to determine the particle size and concentration of exosomes, using the NanoSight NTA 3.4 Analytical software (Malvern NanoSight, Inc.). The measurable particle diameter was 10–40 nm < diameter (d) < 1–2 μm (the lower limit of the size of biological materials [e.g., exosomes] ≈ 60 nm). The particle concentration was ≈10^6^ particles/mL < concentration < 10^9^ particles/mL. The instrument specifications were laser power/wavelength at 40 mW/Red 642 nm and a CCD camera. Additionally, the temperature control range was from 5 °C lower than the ambient temperature to 50 °C.

### 2.5. Western Blot Analysis

Exosomes were lysed using 1X radioimmunoprecipitation assay (RIPA) buffer (40 mM Tris-HCl pH 7.4, 1% Triton X-100, 0.1% sodium dodecyl sulfate [SDS], 0.15 M NaCl, 10% glycerol, 1 mM EDTA, 50 mM sodium fluoride (NaF), 20 mg/mL 1 mM phenylmethylsulfonyl fluoride (PMSF), 1 mM Na3VO4, 5 mM dithiothreitol, 1 μg/mL leupeptin, 1 μg/mL pepstatin, and 1 μg/mL aprotinin) in an extraction solution (Intron Biotech, Inc., Seoul, Republic of Korea). The exosomes were removed by centrifugation at 13,000× *g* for 15 min at 4 °C. Protein concentration was determined by a bicinchoninic acid (BCA) protein assay kit (Pierce, Inc., Rockford, IL, USA).

Each 50 μg protein sample from exosomes was separated by 12% SDS-polyacrylamide gel electrophoresis and transferred to a PVDF membrane. Then, the membrane was washed twice with Tris-buffered saline (pH 7.5, 10 mM Tris, 150 mM NaCl containing 0.1% Tween-20; TBST) and blocked with 5% skim milk in TBST for 1 h at room temperature. Afterward, it was incubated overnight at 4 °C with primary antibodies CD63 (1:1000) and CD81 (1:1000; Abcam, Inc., Cambridge, UK) as a positive marker and Calnexin (1:1000; Abcam, Inc.) as a negative marker. Peroxidase-conjugated anti-rabbit (1:2000; Abcam, Inc.) secondary antibodies were used for 90 min at room temperature and developed with ECL reagent (Santa Cruz, Dallas, TX, USA). Blots reacted with chemiluminescent substrate (ECL; Millipore, Inc.) and were exposed to X-ray film. ImageJ (2.0; National Institute of Health, Bethesda, MD, USA) and GraphPad Prism (version 5.01; GraphPad, La Jolla, CA, USA) were used to count Western blot bands.

### 2.6. VEGFA siRNA-FAM Loading into Exosomes

The isolated exosomes were simply combined with Exo-Fect and selected nucleic acids to create the exosome delivery system to load FAM-labeled VEGFA siRNA (VEGFA siRNA-FAM; shown in green) into exosomes. Briefly, for the chemical loading of exosomes, exosomes were first isolated from the MSC medium by chromatography. Then, the exosome pellet was resuspended in 500 μL of sterile 1× PBS. Afterward, 10 μL Exo-Fect solution (Exo-fect, SBI, Inc., Pittsburg, PA, USA), 20 μL nucleic acid (20 pmol siRNA; Table 1), 70 μL sterile 1× PBS, and 50 μL purified exosomes (1 × 10^6^ particle) were added, followed by placing the transfected exosome sample on ice (or 4 °C) for 30 min. The sample was centrifuged at 14,000 rpm for 3 min in a microfuge. The supernatant was then removed, and the transfected exosome pellet was resuspended in 300 μL.

### 2.7. Transfection of VEGFA siRNA-FAM-Loaded Exosomes into Target Cells

To efficiently deliver VEGFA siRNA-FAM to target cells, the research team utilized the EV-entry system (EV-entry, SBI, Inc.). This system improved the delivery of VEGFA siRNA-FAM by increasing the rate at which exosomes are taken up by recipient cells and by increasing the efficiency at which substances are released within exosomes into the cytoplasm of recipient cells. The EV-entry system consists of components A and B, which are mixed with exosomes just before use. The transfection procedure was as follows: first, the purified exosome pellet was resuspended in 93 μL of sterile DMEM medium (medium without FBS and antibiotics; Gibco, Inc.). To the resuspended exosome suspension (1 × 10^6^ particles), 5 μL of 20X EV-entry reagent A and 2 μL of 1X EV-entry reagent B (EV-entry, SBI, Inc.) were sequentially added. After adding the reagents, the solution was pipetted up and down three times to ensure thorough mixing of the exosomes and reagents. The mixed exosome suspension was then incubated at room temperature for 45 min to allow the reaction to proceed.

### 2.8. RT-PCR Analysis

Total RNAs (1 μg) were used for cDNA synthesis utilizing the First Strand cDNA Synthesis Kit (Thermo Fisher Scientific, Inc., Waltham, MA, USA) to determine the extent of VEGFA mRNA degradation within target cells over time after treatment with VEGFA siRNA-FAM-loaded exosomes. The total RNA concentration was determined using UV-Vis spectroscopy based on absorbance at 260 nm (NanoDrop 2000; Thermo Fisher Scientific, Inc.). For reverse-transcription–polymerase chain reaction (RT-PCR), 1 μg of mRNA was reverse transcribed with oligo (dT) 18 using a thermal cycler (C-1000 Touch; Bio-Rad, Inc., Hercules, CA, USA). PCR was performed using a thermal cycler with reaction mixtures containing Taq DNA polymerase (Thermo Fisher Scientific, Inc.) and appropriate primers (Table 1). Amplification parameters included an initial incubation at 95 °C for 5 min, followed by 38 sequential amplification cycles of 95 °C for 30 s, 55–60 °C for 30 s, and 72 °C for 1 min. The reaction was terminated by incubation at 72 °C for 7 min and then overnight at 4 °C.

### 2.9. FACS Assay

FACS analysis was performed using Annexin V and PI to detect early and late apoptosis when VEGFA siRNA-FAM-loaded exosomes were treated with target cells. Cells from each group were incubated with FITC-conjugated antibodies to annexin V and PE-conjugated antibodies to PI (both provided by Thermo Fisher Scientific, Inc.) for 30 min at room temperature. As controls, cells were stained with FITC-isotype control IgG and PE-isotype control IgG (Thermo Fisher Scientific, Inc.). Then, the cells were washed twice with FACS buffer and analyzed on a FACScan flow cytometer (Becton Dickson, Inc., Franklin Lakes, NJ, USA).

### 2.10. Multiplex Assay

For multiplex immunoassay analysis, the concentrations of several cytokines (VEGFA set, PIGF set, and VEGFC set) obtained from human HUVECs and MSCs were measured using the Bio-Plex^®^ multiplex system (R&D Systems, Inc., Minneapolis, MN, USA) with Luminex-bead-based xMAP technology according to the instructions of the R&D System Manufacturer. Briefly, the research team added 50 µL of the diluted microparticle cocktail to each well and incubated it at room temperature for 2 h on a shaker at 800 rpm. Then, the research team added 50 µL of the standard or sample and 50 µL of the diluted biotin-antibody cocktail to each well. They covered the wells and incubated them for 1 h at room temperature on an 800 rpm shaker. After washing, they added 50 µL of diluted streptavidin-PE to each well and incubated them for 30 min at room temperature in an 800 rpm shaker. After washing again, they removed the liquid from each well, filled it with 100 µL of wash buffer, removed the liquid, and washed it once more. Finally, they added 100 µL of wash buffer to each well and read the results within 90 min using a Luminex^®^ (Thermo Fisher Scientific, Inc.), or Bio-Rad analyzer (Bio-Rad, Inc., Hercules, CA, USA). The concentration of each biomarker was determined by analysis using the Bio-Plex^®^ multiplex system (Bio-Rad, Inc.). All biological fluid samples were diluted at least 2-fold.

### 2.11. Tube Formation Assay

HUVECs (10,000 cells per well) were cultured in a 24-well culture plate (SPL Life Sciences, Inc., Gyeonggi-do, Republic of Korea) pre-coated with Matrigel (1 mL per well; Corning, Inc., One Riverfront Plaza, Corning, NY, USA) to determine whether treatment with VEGF siRNA-FAM-loaded exosomes reduced tube formation. The cells were cultured in a 37 °C incubator for 24 h in a normal culture medium containing FBS (exosome-depleted FBS, Gibco, Inc.). After confirming tube formation, the media was replaced with serum-free media. The next day, the treatment groups were treated with VEGF siRNA-FAM-loaded exosomes (100 μL/mL) according to the treatment time. Tube images were acquired using a camera (Leica Camera AG., Wetzlar, Germany). The tubes in individual wells of the three groups were counted using the ImageJ software program (National Institute of Health). The experiments were performed independently in triplicate.

### 2.12. Confocal Assay

Immunofluorescence staining was performed to determine how exosomes loaded with FAM-labeled VEGFA mRNA (shown in green) appeared in target cells according to time course. The target cells were seeded on 4-well culture plates one day before staining and then treated with drugs the following day. At each time point, the cells were fixed with 4% paraformaldehyde and incubated with a blocking buffer containing 3% FBS and 1% bovine serum albumin (BSA) for 1 h at room temperature. Then, the cells were washed three times with PBS and incubated with DAPI (1:10,000; Invitrogen, Inc., Carlsbad, CA, USA) for 10 min. Fluorescent signals were imaged on a confocal laser scanning microscopy (Carl Zeiss LSM, AG., Oberkochen, Germany) and analyzed using Imaris Viewer software (version 10.0, Oxford Instruments, Inc., Abingdon, UK). All experiments were repeated three times.

### 2.13. Statistical Analysis

All groups had normal distributions and consistent variances. *p*-values * <0.05, ** <0.01, and *** <0.001 indicated statistical significance. All measurements were expressed as mean ± standard error (SE). Statistical significance was calculated using one-way ANOVA, followed by Tukey’s multiple comparison test when comparing different treatments and time points. There were no prior specific inclusion or exclusion criteria.

## 3. Results

### 3.1. Exosome Characterization

The exosome chromatography technique with particle density difference was used to extract exosomes from MSCs. Then, low and high magnifications of exosomes and microvesicles (MVs) were obtained after taking TEM images to confirm whether exosomes were purely extracted (Figure 1A). The average size of exosomes was approximately 30–150 nm, while that of MVs was approximately 150–300 nm. Exosomes showed a hollow shape and were intact without being broken or torn.

The NTA assay was used to measure exosome concentration and size (Figure 1B). Consistent with TEM findings, exosomes and microvesicles (MVs) purified from 300 mL of MSC-conditioned media using chromatography measured approximately 120 ± 1.9 nm and 157 ± 2.7 nm in size, respectively, with concentrations of 1.53 × 10^10^ particles/mL and 1.54 × 10^11^ particles/mL. Western blot was performed to identify exosomes and MVs, using CD63 and CD81 as positive markers and calnexin as a negative marker (Figure 1C). For exosomes, CD63 and CD81 showed overall band signals, while calnexin showed no band.

### 3.2. Inserting VEGFA siRNA-FAM into Exosomes

Exosomes, being essential for cell-to-cell communication, are nanocarriers used by cells to transport RNA and proteins [14]. Exosomes can also deliver exogenous cargo to recipient cells [19]. The research team aimed to increase not only the uptake of exosomes by recipient cells, but also the release of the delivered cargo into the cytoplasm of recipient cells. The transfection effect was measured using siRNA-TX Red to load exosomes into target cells (Appendix A). The red color intensified with increasing exposure time, and the transfection efficiency was particularly better at 48 h after treatment. Hence, hVEGFA siRNA-FAM-EXO (1) and hVEGFA siRNA-FAM-EXO (2) were successfully transfected into 1 × 10^6^ particle exosomes (green color). When the target cells were treated with those, a green signal was observed in the target cytoplasm 24 h after treatment. Additionally, a stronger green signal was observed in the target cytoplasm and around the nucleus at 48 h (Figure 2).

### 3.3. Effect of VEGFA siRNA-FAM on Target Cells

When hVEGFA siRNA-FAM-EXO (1) and hVEGFA siRNA-FAM-EXO (2) were loaded into exosomes and used to treat target cells, the cell morphology according to the treatment time was also confirmed by optical microscopy (Figure 3A). As seen in the confocal image, the FAM green-fluorescent substance attached to hVEGFA siRNA strongly reacted with the target cytoplasm at 24–48 h after treatment. The cell status was similarly confirmed by optical microscopy, which revealed a decrease in cellular length at 24 and 48 h. Overall cell size also diminished as the cells progressed toward death. Figure 3B,C show how much VEGFA mRNA in the target cell was reduced by hVEGFA siRNA according to the exposure time through RT-PCR analysis.

As expected, VEGFA mRNA rapidly decreased at 48 h in both hVEGFA siRNA-FAM-EXO (1) and hVEGFA siRNA-FAM-EXO (2) treatment groups. Specifically, the hVEGFA siRNA-FAM-EXO (1) group showed a significant decrease after 48 h of treatment (0.9 ± 0.07 vs. 0.2 ± 0.03; ** *p* < 0.01). The hVEGFA siRNA-FAM-EXO (2) group showed a significant decrease at 24 h (0.9 ± 0.07 vs. 0.2 ± 0.03; * *p* < 0.05) and 48 h (0.9 ± 0.07 vs. 0.2 ± 0.03; * *p* < 0.05) after treatment.

### 3.4. Cytokine Reduction in MSCs Treated with VEGFA siRNA-FAM-Loaded Exosomes

VEGF levels (VEGFA, VEGFC, and PIGF) produced in the cytosol (lysate) inside the cells and secreted into the CM outside the cells were compared with 0 h when VEGFA siRNA-loaded exosomes were treated with target cells, MSCs, for 30 and 60 h, respectively (Figure 4). Both hVEGFA siRNA-FAM-EXO (1) and hVEGFA siRNA-FAM-EXO (2) showed significantly lower VEGFA at 30 h compared with 0 h, and hVEGFA siRNA-FAM-EXO (2) was also significantly lower at 60 h (* *p* < 0.05 and ** *p* < 0.01, respectively). Regarding CM, both hVEGFA siRNA-FAM-EXO (1) and hVEGFA siRNA-FAM-EXO (2) were significantly lower at 30 and 60 h after treatment than at 0 h (Figure 4A,D; * *p* < 0.05). Furthermore, although the amount of intracellular PIGF did not show any difference over time in both groups, the amount of CM decreased at 30 h after treatment and recovered at 60 h (Figure 4B,E). On the other hand, both hVEGFA siRNA-FAM-EXO (1) and hVEGFA siRNA-FAM-EXO (2) showed higher VEGFC amounts at 60 h after treatment than at 0 h in the internal cytosol, while the CM also showed a similar pattern to the lysate (Figure 4C,F).

### 3.5. Apoptosis Induced by VEGFA siRNA-Loaded Exosomes

To confirm whether the cells were actually dead, flow cytometry using Annexin V-FITC (green) and PI-PE (red) was performed (Figure 5A,B). Both hVEGFA siRNA-FAM-EXO (1) and (2) treatment groups showed increased cell death over time compared to the untreated group (0 h). In Figure 5B, the percentage of viable cells decreased from 100 to 84.0 ± 2.5% (0.84-fold; * *p* < 0.05) in group (1) and from 100 to 84.7 ± 4.0% (0.85-fold; * *p* < 0.05) in group (2) at 24 h, and further decreased to 38.4 ± 3.0% (0.38-fold; *** *p* < 0.001) in group (1) and 65.2 ± 2.7% (0.65-fold; ** *p* < 0.01) in group (2) at 48 h. On the other hand, at 48 h, early apoptosis was significantly increased to 20.1 ± 3.8% (* *p* < 0.05) in group (1) and 14.5 ± 5.6% (* *p* < 0.05) in group (2), and late apoptosis was significantly increased to 36.4 ± 6.5% (** *p* < 0.01) in group (1) and 14.8 ± 2.5% (* *p* < 0.05) in group (2). However, since 0 h was 0%, the fold change for the corresponding item was not calculated. No significant change in necrosis was observed over time.

### 3.6. Tube Formation Effect

To evaluate the effect of hVEGFA siRNA-loaded exosomes (hVEGFA siRNA-FAM-EXO) on in vitro angiogenesis, the research team conducted an experiment to induce HUVEC tube formation (Figure 6A,B). Initial analysis results revealed no significant difference in tube formation ability between the control group and the hVEGFA siRNA-FAM-EXO (1) and (2) treatment groups up to 24 h after treatment. However, from 48 h, angiogenesis in the hVEGFA siRNA-FAM-EXO (2) treatment group began to significantly decline, decreasing approximately 2.7-fold (365.3 ± 64.1 vs. 135.6 ± 24.2) compared to the control group. In particular, at the 72 h time point, angiogenesis was almost completely inhibited, decreasing approximately 11.3-fold (300.3 ± 108 vs. 26.6 ± 5.8) compared to the control group. The hVEGFA siRNA-FAM-EXO (1) treatment group also demonstrated a similar trend. After 72 h of treatment, angiogenesis was significantly reduced, decreasing approximately 4.6-fold (300.3 ± 108 vs. 64.7 ± 38.7) compared to the control group. Confocal microscopy images of cell morphology and status clearly revealed changes according to treatment time when hVEGFA siRNA-FAM-EXO (1) and (2) were administered to tube-forming cells (Figure 6C). After 72 h of treatment, FAM green-fluorescent material attached to hVEGFA siRNA was observed to strongly react with target cells. In addition, tube size decreased at 48 h compared to 24 h and gradually disappeared thereafter in both treatment groups.

### 3.7. Reduction in Cytokines (VEGFA, PIGF, VEGFC) in HUVECs Treated with VEGFA siRNA-FAM-Loaded Exosomes

Exosomes loaded with VEGFA siRNA showed antiangiogenic effects in the in vitro angiogenesis assay, confirming differences in VEGFA, PIGF, and VEGFC levels according to the treatment time in HUVECs (Figure 7). First, VEGFA (lysate) amounts in the cell cytosol of HUVECs in the hVEGFA siRNA-FAM-EXO (1)- and hVEGFA siRNA-FAM-EXO (2)-treated groups significantly decreased after 30 h after treatment compared to 0 h and increased at 60 h. However, the protein amounts in the CM secreted from the cells significantly decreased at both 30 and 60 h (*** *p* < 0.001) (Figure 7A,D).

Both PIGF amount inside the cytosol of HUVECs and in the secreted CM outside the cell significantly decreased over time in both hVEGFA siRNA-FAM-EXO (1)- and (2)-treated groups (* *p* < 0.05; ** *p* < 0.01; *** *p* < 0.001; Figure 7B,E). Meanwhile, the secretion pattern of VEGFC inside the HUVEC cytosol showed a similar pattern to that of PIGF secretion: the amount of intracellular protein (lysate) decreased over time in hVEGFA siRNA-FAM-EXO (1)- and hVEGFA siRNA-FAM-EXO (2)-treated groups. Both groups showed similar patterns, and in the case of CM, unlike lysate, it decreased at 30 h and increased at 60 h after treatment (Figure 7C,F).

## 4. Discussion

Ocular diseases caused by or associated with neovascularization include corneal neovascularization, neovascular glaucoma, and diabetic retinopathy [5]. Particularly, age-related macular degeneration (AMD) is the most common cause of aging-related vision loss and is mainly characterized by the degeneration of the macula, the central part of the retina [29]. VEGFA plays an important role in the pathophysiology of macular degeneration [30]. In fact, VEGFA overexpression is one of the main causes of macular degeneration, leading to new blood vessel formation and visual acuity loss [29]. Therefore, VEGFA inhibition is considered a major strategy for treating macular degeneration [31].

However, existing anti-VEGF treatments have several limitations [32]. For example, anti-VEGF protein injections have limited therapeutic effects and require repeated administration, which is inconvenient. Furthermore, VEGF siRNA is effective in directly inhibiting VEGF but has low stability and delivery efficiency [33]. Therefore, exosome-based VEGFA siRNA delivery technology has recently attracted attention. Exosomes have the advantages of high in vivo stability and excellent intracellular delivery efficiency. Additionally, loading VEGFA siRNA into exosomes not only improves targeting to specific cells or tissues but also has the potential to improve therapeutic effects [34]. Therefore, developing a new VEGFA inhibition strategy for treating macular degeneration is important. Accordingly, optimization of the exosome-based VEGFA siRNA delivery system and verification of its therapeutic effects are necessary [35].

Related research and development of exosome-based VEGFA siRNA delivery are garnering attention as a promising approach that can overcome the limitations of existing treatments and are expected to provide new possibilities for the treatment of macular degeneration. Our research team tested the effect of drug delivery by inserting FAM-tagged VEGFA siRNA into exosomes to effectively remove neovascularization in vitro. First, the research team extracted exosomes from MSCs and analyzed their size and morphology using NTA and TEM to confirm the extraction and characterization (Figure 1). Specifically, the team extracted exosomes using a particle size-based column fractionation method, utilizing a conventional chromatographic technique that achieves higher purity while saving time [28]. Furthermore, to confirm the high loading efficiency, the lipofectamine method with an immunofluorescence assay was used for siRNA-Texas Red-loaded exosomes, showing high siRNA-TX expression (Appendix A) [36]. Based on the above-mentioned conditions, VEGFA siRNA was loaded into exosomes using the lipofectamine method to load FAM labeled with the VEGFA siRNA sequence; this was followed by measuring VEGFA siRNA using an immunoaffectivity assay [37,38].

Successful introduction of hVEGFAsiRNA-FAM into target cells was achieved with high efficacy, especially after 48 h of treatment. This was demonstrated by an approximately 5.0-fold (400%) increase in introduction for hVEGFAsiRNA-FAM (1) and an approximately 4.0-fold (300%) increase in introduction for hVEGFAsiRNA-FAM (2) (Figure 2 and Figure 3). In the target cells, the cellular uptake of VEGFA siRNA-loaded exosomes and the increased effect of suppressing VEGFA expression were confirmed (Figure 3). RT-PCR analysis confirmed that VEGFA mRNA was effectively suppressed 48 h after treatment. VEGFA siRNA-loaded exosomes entered the target cells (MSCs) and degraded VEGFA mRNA. Using a multiplex assay, the research team determined the amount of VEGFA protein produced intracellularly and secreted extracellularly over time in MSCs (Figure 4). According to Bayraktar et al., microRNAs (miRNAs) delivered via exosomes might play an important role in cell-to-cell communication by mediating the suppression of important mRNA targets in neighboring or more distant recipient cells [30].

As a result, the research team confirmed differences in the levels of various VEGFs (VEGFA, PIGF, VEGFC) secreted from exosomes loaded with VEGF siRNA depending on the treatment time. Additionally, the research team confirmed a significant decrease in cytoplasmic VEGFA levels, as well as in CM secreted from MSCs. On the other hand, PIGF and VEGFC did not show significant differences. In particular, VEGFC showed an increase when treated with VEGF siRNA (Figure 4). PIGF in MSCs is mainly involved in immune regulation and tissue regeneration [39]. According to Park et al., the biological activity of PlGF secreted from the co-culture supernatant was functional, as evidenced by the increased angiogenesis and chemotaxis of co-cultured CM-induced MSCs, both of which were significantly inhibited by anti-PlGF antibodies. These data highlight the importance of MSC/FLS interaction in increased PlGF production, confirming the angiogenic activity of PlGF. Additionally, VEGFC has a similar function to VEGFA as a growth factor mainly involved in lymphangiogenesis. Cursiefen et al. showed that VEGFA is chemotactic for macrophages, which release lymphangiogenic VEGFC/VEGFD in inflamed corneas. Furthermore, they evaluated the possibility that macrophage recruitment plays a role in VEGFA-mediated lymphangiogenesis. When VEGF expression is decreased by VEGF siRNA treatment, VEGFC expression can be increased [40]. This is because VEGF plays a role in suppressing VEGFC expression. Therefore, using VEGF siRNA is likely to increase VEGFC expression [41]. Wang et el. showed that when VEGFA siRNA was injected into BTT-T739 tumors, the volume of the ipsilateral inguinal lymph nodes increased compared to the control group, which was morphologically confirmed as reactive proliferation [41]. VEGFA was significantly reduced in the siVEGFA group, demonstrating that siVEGFA efficiently suppresses VEGFA expression in tumors. VEGFR2, a VEGFA receptor that partially indicates vascular density, was also downregulated in both siRNA treatment groups. VEGFC and VEGFR3 expression did not significantly change in siRNA treatment groups.

The FACS assay was performed to examine MSC apoptosis caused by VEGF siRNA, confirming that early and late apoptosis increased 48 h after treatment (Figure 5). Wiszniak et al. found that endocrine VEGF is also involved in endothelial cell homeostasis and survival, showing that endothelial-specific VEGF knockout in VE-CAD-Cre; Vegfafl/fl mice resulted in progressive endothelial degeneration in adults in vivo, while apoptosis increased in the in vitro culture of VEGF-deficient endothelial cells [42]. Similarly, it increased the severity of systemic vascular defects observed by Lee et al. [43], suggesting that cellular stress affected the phenotype. In detail, stress induced by radiation, reactive oxygen species, and hypoxia triggers VEGFR2 activation by both autocrine and endocrine VEGF sources, which support endothelial survival. Without endocrine VEGF, some endothelial cells undergo apoptosis, leading to hemorrhage (small vessels), exposure of the basement membrane, and thrombosis (large vessels). VEGF knockdown by siRNA in HUVECs resulted in increased cell death, which was not reversed by adding recombinant VEGF [42].

Meanwhile, VEGFA forms a complex with VEGFR-2 in endothelial cells and may be involved in the same complex that includes VEGF-A, VEGFR-2, and EEA1, maintaining the basal activation level of VEGFR-2-regulated signaling in most cells [44]. These components may enhance FoxC2 nuclear translocation or alter FoxC2 binding to FOX:ETS motifs in various endothelial markers. When VEGFA was knocked out, cell death was induced alongside the disruption of cellular homeostasis.

Upon VEGFA knockout, tube formation progressively decreased over time. In particular, the hVEGFA siRNA-FAM (1)-treated group (20 segments; total segment length: 1000) and hVEGFA siRNA-FAM (2)-treated group (50 segments; total segment length: 1200) exhibited significantly reduced tube formation at 72 h post-treatment compared to the untreated group (Figure 6A,B). For reference, the number of segments and total segment length were used as quantitative indicators of tube formation, providing insights into both the functional and structural characteristics of the tubes [45]. Additionally, the research team localized FAM-tagged VEGFA siRNA-exosomes (imaged in green) to determine whether VEGFA siRNA inhibited tube formation over time (Figure 6C).

Figure 7 shows the multiplex assay results for the amount of VEGFA siRNA, VEGFA, and PIGF secreted by HUVECs—endothelial cells involved in tube formation—in both the intracellular cytosol and CM. When the VEGFA siRNA was treated, unlike in the CM group, VEGFA decreased at 30 h after treatment and then slightly increased at 60 h. Considering its nature, siRNA can enter the target cells, HUVECs, in a packed state in exosomes, provide its effects while having a short degradation half-life, and recover at 60 h after treatment [46]. Conversely, in the case of CM, VEGFA secreted in the medium was concentrated, decreasing over time. In the case of VEGFC and PIGF, a similar pattern was observed with VEGFA. Particularly, PIGF is a factor contributing to angiogenesis by interacting with VEGF, and the expression of PIGF is similar after VEGF siRNA treatment. The difference between VEGFA and PIGF in the cytosol group was that VEGFA strongly binds to and activates VEGFR-1 and VEGFR-2. On the other hand, PIGF is mainly activated by VEGFR-1. Hence, the difference in VEGFA siRNA function is related to the difference in their activity [47].

Guangqi et al., similarly to our study, also evaluated the effect of endogenous VEGF-A knockdown on endothelial cell morphogenesis in the extracellular matrix [48]. They found that while control siRNA-treated cells formed tubular structures very rapidly after seeding cells on Matrigel, endogenous VEGFA knockdown clearly inhibited tube formation in HUVECs. Interestingly, endothelial cell membrane VE-cadherin was significantly attenuated in endogenous VEGFA knockdown cells compared with control siRNA-treated cells. Of note, VE-cadherin is an endothelial cell-specific adhesion molecule located at the junction between endothelial cells and is required for vascular remodeling and morphogenesis [49].

This study demonstrates that exosome-based delivery strategies offer superior safety compared to viral vector systems, primarily due to their low immunogenicity and risk of genome integration [16]. Additionally, they effectively overcome the limitations of siRNA itself [50], thereby protecting siRNA and maximizing intracellular delivery efficiency. These findings demonstrate the potential of exosome-based systems to overcome major limitations of previous siRNA delivery approaches. Although challenges related to the mass production and standardization of exosomes persist [15], this study, as a follow-up to our previous work [15] and an extended validation of our work on biomimetic carriers [51], makes significant progress. In particular, it addresses the limitations of mass production and standardization of exosome purification protocols by applying chromatographic purification techniques. Direct observations using FAM-labeled siRNA and specific in vitro effects—such as reduction in VEGFA, PIGF, and VEGFC—and induction of apoptosis, suggest high siRNA loading and delivery efficiency within exosomes. These findings represent a significant advance in optimizing siRNA loading techniques.

To advance this exosome-based VEGFA siRNA delivery strategy toward successful clinical application, several critical challenges, including the following, must be resolved: standardization of exosome isolation/modification technology, efficient loading and precise dose control of VEGFA siRNA, securing mass production scalability, and long-term safety verification. If these challenges are resolved, this strategy is expected to provide an innovative solution for the treatment of various angiogenic diseases and contribute substantially to the improvement of human health.

## 5. Conclusions

This study clearly demonstrated that exosomes loaded with FAM-tagged VEGFA siRNA effectively reduced VEGFA, PIGF, and VEGFC expression and induced apoptosis in target cells. These findings support the notion of a stable intracellular delivery of VEGFA siRNA via exosomes and an effective silencing mechanism of abnormal angiogenesis-related genes. These results suggest a promising therapeutic strategy for various angiogenic diseases, including neovascular ocular diseases such as keratitis, neovascular glaucoma, and diabetic retinopathy models. In particular, considering the limitations of existing anti-VEGF therapies, exosome-based VEGFA siRNA delivery has the potential to provide a new therapeutic alternative with more efficacy and fewer side effects. However, for successful clinical applications in the future, several challenges must be addressed, including standardization of exosome isolation and modification techniques, efficient loading and precise therapeutic dose control of VEGFA siRNA, scalability for large-scale production, and long-term safety verification for clinical applications. If these challenges are successfully addressed, the exosome-based VEGFA siRNA delivery strategy is expected to provide an innovative solution for the treatment of various angiogenic diseases and have a practical impact on human health.

## Figures and Tables

**Figure 1 bioengineering-12-00919-f001:**
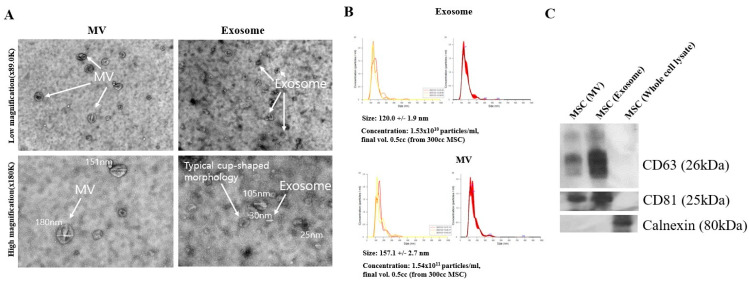
Characterization of exosomes. Exosomes secreted from MSCs were purified using a chromatographic exosome isolation method, and the presence of exosomes was confirmed by TEM images (**A**), nanoparticle tracking analysis (NTA; **B**), and Western blot analysis (**C**). MSC (MV)—microvesicles isolated from MSCs; MSC (exosomes)—pure exosomes isolated from MSCs; MSC (whole cell lysate)—whole cell lysate isolated from MSCs.

**Figure 2 bioengineering-12-00919-f002:**
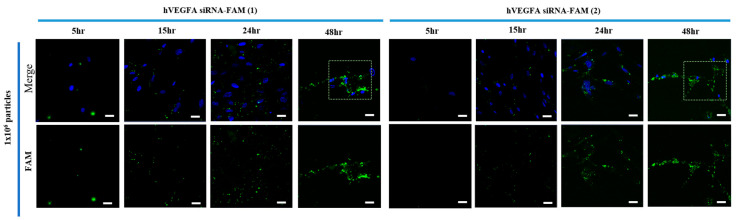
Exosome insertion into MSCs. VEGFA siRNA-FAM (green fluorescent) was successfully loaded into exosomes and delivered to target cells. Green dotted frame indicated fluorescently labeled exosomes with VEGFA siRNA-FAM. FAM-labeled VEGFA siRNA (green, confirming cytoplasmic reach/distribution within target cells) & DAPI (blue, nuclear staining) were stained as above. The scale bar indicates 200 μm.

**Figure 3 bioengineering-12-00919-f003:**
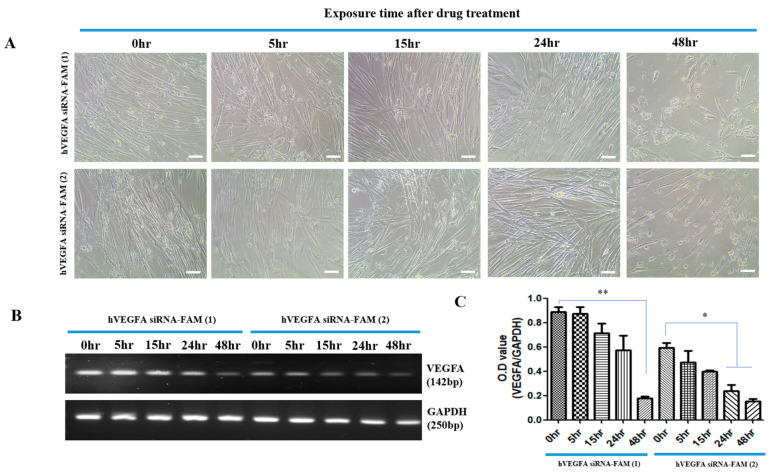
VEGFA siRNA-FAM effect on target cells. Images of morphological changes in MSCs over time after treatment with exosomes loaded with VEGFAsiRNA-FAM (**A**). Images of the decrease in VEGFA mRNA in MSCs using electrophoresis (**B**) and graph (**C**). *p* values of * <0.05 or ** <0.01 were considered statistically significant. The scale bar represents 100 μm.

**Figure 4 bioengineering-12-00919-f004:**
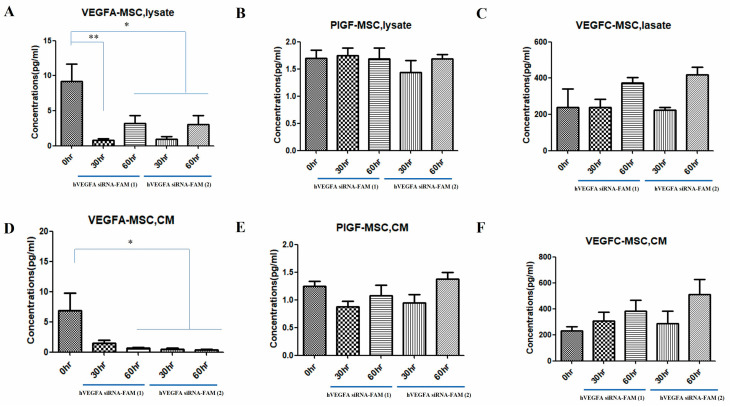
Reduction in cytokines (VEGFA, PIGF, VEGFC) in target MSCs by treatment with exosomes loaded with VEGFA siRNA-FAM. The graph showed the quantification of VEGFA, PIGF, and VEGFC protein levels in the intracellular cytosol and conditioned medium (CM) secreted from cells by the effect of VEGF siRNA using multiple tests. Multiplex assay for VEGFA, lysate (**A)**, PIGF, lysate (**B**), VEGFC, lysate (**C**), VEGFA, CM (**D**), PIGF, CM (**E**), VEGFC, CM (**F**) from MSCs. *p* values of * <0.05 or ** <0.01 were considered statistically significant.

**Figure 5 bioengineering-12-00919-f005:**
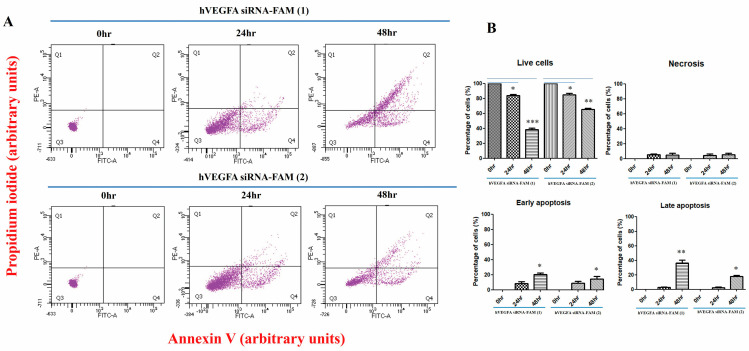
Apoptosis induced by exosomes loaded with VEGFA siRNA-FAM. The effect of VEGFA siRNA over time was confirmed by FACS analysis (**A**) and graphs (**B**). *p* values of * <0.05, ** <0.01, *** <0.001 were considered statistically significant.

**Figure 6 bioengineering-12-00919-f006:**
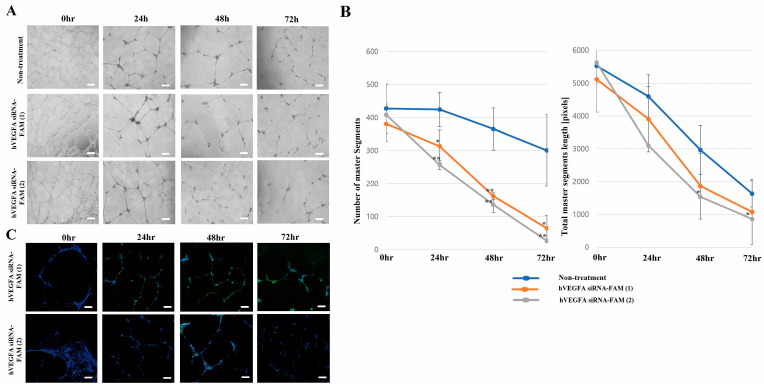
Effect of VEGFA siRNA-FAM under tube formation. Morphological images of tube formation from HUVECs over time after treatment with VEGFA siRNA-FAM-loaded exosomes (**A**). Graph of the number of master segments involved in tube formation and the total master segment length (**B**) and the confocal images (**C**). *p* values of * <0.05, or ** <0.01 were considered statistically significant. The scale bar indicates 200 μm.

**Figure 7 bioengineering-12-00919-f007:**
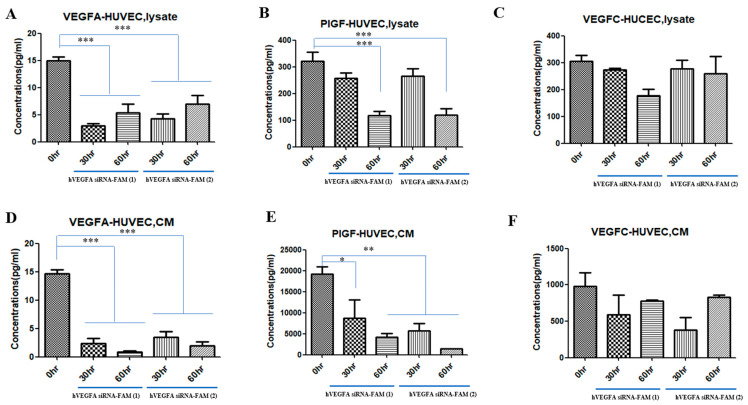
Reduction in cytokines (VEGFA, PIGF, VEGFC) in HUVEC by treatment with exosomes loaded with VEGFA siRNA-FAM. The graph shows the quantification of VEGFA, PIGF, and VEGFC protein levels in the intracellular cytosol and conditioned medium (CM) secreted from cells by the effect of VEGF siRNA using multiple tests. Multiplex assay for VEGFA, lysate (**A**), PIGF, lysate (**B**), VEGFC, lysate (**C**), VEGFA, CM (**D**), PIGF, CM (**E**), VEGFC, CM (**F**) from HUVEC. *p* values of * <0.05, ** <0.01, or *** <0.001 were considered statistically significant.

**Table 1 bioengineering-12-00919-t001:** The information for VEGFA siRNA sequences and VEGFA primer.

siRNA	Sequences (5′-3′)	
VEGFA siRNA#1	Sense: FAM-CUG AUA CAG AAC GAU CGA U=tt(1-AS)	
	Antisense: AUC GAU CGU UCU GUA UCA G=tt(1-AA)	
VEGFA siRNA#2	Sense: FAM-CAG AAC GAU CGA UAC AGA A=tt(2-AS)	
	Antisense: UUC UGU AUC GAU CGU UCU G=tt(2-AA)	
Primer	Sequences (5′-3′)	Product size
VEGFA	Sense: CAGTTCGAGGAAAGGGAAAGG	
	Antisense: CAACGCGAGTCTGTGTTTTTG	160 bp

## Data Availability

The original contributions presented in the study are included in the article/Appendix A, further inquiries can be directed to the corresponding authors.

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
