# Peer review of "The Antiangiogenic Effect of VEGF-A siRNA-FAM-Loaded Exosomes"

_bioengineering, 2025, doi:10.3390/bioengineering12090919_

Round 1

Reviewer 1 Report

Comments and Suggestions for Authors

Woojune Hur and colleagues have conducted a study proposing a delivery strategy using VEGFA siRNA and exosomes, supported by FAM labeling for in vitro visualization. As the manuscript primarily focuses on in vitro research, which is indeed solid, I have only the following three comments:

  1. For Figure 1 and Figure S1, please use green dashed boxes to indicate highlighted regions instead of orange arrows. In Figure 7A and 7D, it is unclear what the authors intend to convey with the double line asterisks (*). Please clarify.

  1. In fact, many existing publications have already reported similar delivery strategies, including those based on VEGF siRNA (PMID: 18767037, 18978955) and extracellular vesicles (PMID: 38802884, 39111599) . However, this manuscript focuses almost exclusively on the authors’ own work and largely overlooks prior studies. Please include a discussion of these and other potentially relevant studies in the manuscript.

  1. While Figure 1 presents TEM images of exosomes, it appears that the VEGFA siRNA-FAM–loaded exosomes were not imaged or shown. Including such an image could enhance the characterization of the loaded complex.

Author Response

We would submit the response to the reviewer 1's report as the attached file. 

Reviewer 2 Report

Comments and Suggestions for Authors

Comments to Authors:

The authors present an interesting study investigating the antiangiogenic effects of exosomes and the delivery of VEGF-A siRNA-FAM. The work offers valuable insights into the potential of exosome-mediated VEGF siRNA delivery as a less invasive therapeutic approach for ocular disorders such as diabetic retinopathy and macular degeneration, which currently rely on invasive treatment strategies to manage neovascularization.

While the manuscript highlights important findings, major revisions are recommended to improve grammar, enhance sentence clarity, and potentially strengthen the Introduction and Materials and Methods sections.

Please find detailed comments below:

Major Comments:

  1. Grammar and Clarity:

While the manuscript is generally well written, a thorough grammatical review and refinement of sentence structure are recommended to enhance readability and scientific clarity.

  1. Introduction:
  1. There is noticeable repetition in this section, where the same ideas are conveyed multiple times. For instance, the role of exosomes in carrying proteins, lipids, and RNA is mentioned in both lines 55–57 and again in lines 70–72. Similarly, the function of exosomes in signal transmission and material transport appears in both lines 57–59 and 63–65. To maintain scientific clarity and improve the flow of the introduction, it is recommended that the authors either merge similar points or expand on additional functions of exosomes relevant to drug delivery. Providing a broader functional context would strengthen the introduction and offer more depth to the rationale behind using exosomes.
  2. The introduction concludes with the statement that exosomes may offer greater safety and efficacy compared to current treatments. This point would be more compelling if the authors briefly outlined the existing standard treatments for ocular neovascular disorders and highlighted their limitations. Such context would position exosomes as a potentially superior alternative and strengthen the rationale for this study.

3. Exosome purification:

    1. Refining the grammar and rephrasing certain sentences in the Methods section would enhance scientific clarity and make the exosome purification process easier to understand.
    2. It would be helpful for the authors to clarify how fractions 11–13 were distinguished or separated. Additionally, please elaborate on the rationale for selecting fractions 11 and 12 as exosome-containing fractions. Was this decision based on specific markers, size distribution, density, or any other method of characterization?

4. Several abbreviations throughout the manuscript are not defined upon first use, such as PMSF, NaF, BCA, and MEM. For clarity and to maintain the flow of reading, please ensure that all abbreviations are clearly defined when first introduced in the text.

5. VEGFA siRNA-FAM loading into exosomes:

Please ensure that the description of experimental procedures is complete and clear. The sentence on lines 157–158 appears incomplete. Specifically, it is unclear what vehicle (300 µL) was used to resuspend the pellet.

  1. Multiplex assay:

The manuscript states that 50 L of diluted microparticle cocktail was added to the well. This appears to be a typographical error, as 50 liters is not a practical volume for this context. The intended unit is likely µL (microliters). Please correct the unit accordingly to ensure accuracy.

Thank you and best of luck.

Comments on the Quality of English Language

Some sections of the manuscript are written in the first-person voice, and several sentences lack scientific precision and grammatical clarity. A thorough revision focusing on grammar and paraphrasing would greatly enhance the manuscript’s overall quality and readability. 

Author Response

We would submit the response to the reviewer 2's report as the attached file. 

Reviewer 3 Report

Comments and Suggestions for Authors

The manuscript entitled “The antiangiogenic effect of VEGF-A siRNA-FAM–loaded exosomes”  by Jae Yong Kim and et al. is an interesting research, well written and well planned. The topic is original and highly relevant to the field of neovascular ocular diseases are caused by vascular endothelial growth factor A (VEGFA) overexpression research. Finding better treatments for this disease is a clinical priority. VFGFA is known to modulate neovascular processes.

  1. The introduction clearly states the problem: neovascular ocular diseases driven by VEGFA overexpression, but the authors could wrote more about limitations of current anti-VEGF therapies.
  2. What is context on existing anti-VEGF therapies. It would help to briefly name or describe examples of current therapies (e.g., ranibizumab, bevacizumab) and mention the mode of administration (intravitreal injections), which are known to have compliance and safety issues.
  3. Accuracy of the experimental design, methods and statistical analysis is well prepared and project. The methodology is well-written and includes many details that are important for its replicability and/or reproducibility.
  4. Conclusion are consistent with evidence, but in my opinion they could be discussed much more insightful with particular attention to the mechanism and the translation to people. I thing this method could be challenging. These challenges can be related to standardization of exosome isolation and modification, control of therapeutic load, scalability and safety in clinical applications.
  5. In my opinion all figures are clear and well described. They properly show the data and are easy to interpret.
  6. The authors did not describe the limitations of their study. Therefore, it would be valuable to present some perspectives for future research.

Author Response

We would submit the response to the reviewer 3's report as the attached file. 

Reviewer 4 Report

Comments and Suggestions for Authors

Dear Authors,

Thank you for the opportunity to review your interesting manuscript titled "Exosome-loaded VEGFA siRNA inhibiting angiogenesis." The study offers valuable insights into VEGFA inhibition pathways and the use of siRNA-loaded exosomes. However, the manuscript would benefit from substantial revisions, particularly in terms of presentation, grammar, and overall clarity.

In its current form, the Results and Interpretation including the Discussion sections are difficult to follow. Additionally, the frequent reference to siRNA 1 and siRNA 2 throughout the manuscript suggests a comparative analysis of the two siRNA constructs and their efficacy. While this is informative, the study would be significantly strengthened by comparing siRNA-based approaches with alternative strategies for VEGFA inhibition, which could provide a broader context and greater impact.

I have included specific comments and suggestions within the attached manuscript to guide your revisions.

Comments on the Quality of English Language

Many paragraphs within the manuscript are not understandable, and to make it clearer, more scientifically polished, and grammatically correct, the manuscript needs a substantial revision throughout. 

Author Response

We would submit the response to the reviewer 4's report as the attached file. 

Round 2

Reviewer 1 Report

Comments and Suggestions for Authors

Thank you to the authors for the revisions. I believe this work has now reached an acceptable standard, and I have conveyed my recommendation to the editor to accept the manuscript.

However, I would like to briefly outline my remaining concern, which mainly focuses on Q3. I raised this question because part of the manuscript’s title refers to “VEGF-A siRNA-FAM–loaded exosomes.” However, with regard to the characterization of this material, the authors may not have provided sufficiently direct evidence. I certainly acknowledge the further explanations provided in the responses, but it seems that the absence of such direct data somewhat weakens the authors’ efforts, even though the work, as a whole, tells a very complete story.